# Distinct Fecal Proteolytic Activity in Zoo Animals with Different Feeding Strategies

**DOI:** 10.3390/ani15243559

**Published:** 2025-12-11

**Authors:** Luka Otte, Arryn Baltus, Floris J. Bikker, Anouk Fens, Kamran Nazmi, Heleen van Engeldorp Gastelaars, Henk S. Brand, Wendy E. Kaman

**Affiliations:** 1Department of Oral Biochemistry, Academic Centre for Dentistry Amsterdam, University of Amsterdam and VU University Amsterdam, Gustav Mahlerlaan 3004, 1081 LA Amsterdam, The Netherlandsf.bikker@acta.nl (F.J.B.); k.nazmi@acta.nl (K.N.); h.brand@acta.nl (H.S.B.); 2Dierenpark Amersfoort, 3819 AC Amersfoort, The Netherlands; afens@dierenparkamersfoort.nl (A.F.); hpost@dierenparkamersfoort.nl (H.v.E.G.)

**Keywords:** feces, protease, carnivore, herbivore, omnivore, feeding strategy

## Abstract

The type of food animals eat—meat, plants, or both—may influence how their gut breaks down proteins. To unravel the link between diet and protein-digesting enzymes we measured the natural activity of these enzymes in the feces of carnivores, omnivores, and herbivores. Fecal samples were tested using a fluorescent method to visualize enzyme activity, and enzyme blockers were added to identify which enzymes were most active. Carnivores and omnivores showed much higher enzyme activity than herbivores. These results suggest that diet strongly affects protein-digesting enzyme activity in the gut, meaning an animal’s food directly influences its digestive function.

## 1. Introduction

Proteases are enzymes which cleave proteins and can be classified into different families based on the mechanism of hydrolytic cleavage including serine proteases and metalloproteases [1]. They play a role in various processes such as cell proliferation, tissue remodeling, blood clot formation, wound healing, immune response and food digestion [2]. Compared to other organs, the gastrointestinal tract contains the highest levels of proteases [2,3]. Disbalance of this proteolytic activity contributes to epithelial damage and increased intestinal permeability, which plays a role in the pathophysiology of various intestinal diseases such as gastroenteritis and inflammatory bowel disease (IBD) [4,5].

A currently investigated therapeutic strategy for IBD targets this imbalance in proteolytic activity through the use of (serine) protease inhibitors [6]. It was, for example, shown that mice with IBD treated with protease inhibitors (e.g., nafamostat or elafin) have reduced inflammation and a restored mucosal barrier function [7,8]. However, most protease inhibitors are protein-based molecules which gives them poor oral bioavailability because they are readily degraded within the gastrointestinal tract [9].

Another method could be to adjust the patients’ diet. In this study we aimed to investigate whether the gut protease disbalance can be influenced, and potentially restored, by diet. Animal- and plant-based proteins differ in their digestibility, and some animal proteins are more bioavailable, making them easier to digest [10]. For this reason, we hypothesize that fecal proteolytic activity might relate to the source of the proteins present in the diet. This is strengthened by the observation that mice fed egg white have a higher protease abundance in their feces than mice fed soybeans, rice or peas [11]. To confirm this result in a more natural setting we compared fecal proteolytic activity between different zoo animals with varying dietary preferences.

The animals selected for this research have evolved distinct adaptations to consume and digest proteins, reflecting the diverse natural diets they consume. For example, large carnivores, like the lion and tiger, feed mainly on meat, which is rich in animal proteins. To break down these proteins efficiently, they produce high-level concentrations of digestive proteases such as pepsin, trypsin, and chymotrypsin [12,13]. Insectivores are adapted to their diet by producing specific enzymes which digest chitin, the main component of insect exoskeletons. Examples of such enzymes are chitinases and some lysozymes [14]. Piscivores, including otters, primarily eat fish, which are rich in fats like omega-3 oils [15]. Therefore, this group of animals potentially has a high activity of pancreatic lipase and bile salt-activated lipase to manage the high oil content of their diet. Herbivores obtain their proteins from plant materials, which are less digestible due to plant cell walls and anti-nutritional factors like trypsin inhibitors and lectins [16]. Additionally, since plant proteins are often bound in fibrous matrices, herbivores digest some protein via host proteases, but a large fraction comes from microbial protein and microbial enzymes in the gut [17]. Due to the mentioned low dietary protein and their reliance on microbial proteolysis the secretion of digestive proteases is potentially lower compared to carnivores [18].

Lastly, omnivores have adapted to a mixed diet with both animal and plant proteins. Most omnivores are primarily herbivores that consume animal protein during specific times of the year with the ratio of animal-to-plant protein intake varying significantly between species [19]. Because of this mix of animal and plant protein consumption, the digestive system of omnivores is versatile. This allows them to produce a balanced range of proteases to effectively digest various protein sources, making them well-suited to a diverse diet.

Thus, the protease production of each animal is matched to the source and type of proteins in their natural diet. This implicates an intricate relationship between digestive protease activity in feces and diet.

## 2. Materials and Methods

### 2.1. Collection of Fecal Samples

Samples were collected from herbivorous, carnivorous, and omnivorous zoo animals at Dierenpark Amersfoort (Amersfoort, The Netherlands) and Diergaarde Blijdorp (Rotterdam, The Netherlands). The health status of all zoo animals used was monitored on a regular basis by professional caretakers and veterinarians employed by the zoos. Daily fresh fecal samples were collected by the caretakers and pooled from three different locations within the enclosure. Feces was collected on three or four consecutive days resulting in three to four samples per enclosure. The collected samples are thus mixed from all the animals present within the enclosure. The herbivore group consisted of three sub-groups: species mainly eating fruits (frugivore) (*n* = 2), fore-gut fermenters (ruminants) (*n* = 2) and hind-gut fermenters (non-ruminants) (*n* = 2). The carnivore group consisted of three sub-groups: species eating mainly meat (carnivores) (*n* = 2), species eating mainly insects (insectivores) (*n* = 2) and species eating mainly fish (piscivores) (*n* = 2). Selection of the animal groups was based on the natural feeding strategy rather than the captive diet and only samples from healthy animals were included. Upon collection, fecal samples were immediately stored at −20 °C. Numerous (>3) freeze–thaw cycles were avoided since it affects proteolytic activity. Type and amount of food, dietary preference and number of animals present in the enclosure were documented. An approximate host phylogeny tree was created using the TimeTree database based on the species names involved [20].

### 2.2. Ethical Statement

The research was performed in full accordance with the EAZA Code of Ethics (EAZA, 2015) and the Ethical Guidelines for the Conduct of Research on Animals by Zoos and Aquariums (WAZA, 2005), complied with European Directive 2010/63/EU and its implementation in Dutch legislation through the Dutch Animal Experimentation Act. The study did not require approval by a Dutch Animal Experiments Committee, because all procedures were strictly non-invasive and caused no pain, suffering, distress, or harm to the animals, with all samples being conducted under an ethical agreement with the hosting zoological institutions to ensure high standards of welfare and minimal disturbance to natural behavior, and remained fully compliant with applicable national legislation and international best practices. All animals from which fecal samples were collected in this study were housed in accordance with relevant guidelines and regulations.

### 2.3. Proteolytic Profiling

To screen for proteolytic activity in fecal samples, 21 fluorescence resonance energy transfer (FRET) peptide substrates were used (Table 1). All substrates contained two amino acids flanked by a 6-aminohexanoic acid (Ahx)-coupled fluorescent probe (fluorescein isothiocyanate (FITC)) and a lysine-coupled quencher (Dabcyl) [21].

Fecal samples were first dissolved in 20 mM Hepes buffer containing 0.05% Tween20 (pH 8.2) at a concentration of 10 mg/ mL. Of these suspensions, 49 µL was added to 1 µL of 800 µM FRET-peptide substrate in a black, clear-bottom 96-well plate (Greiner Bio-One, Alphen a/d Rijn, The Netherlands). The negative control contained feces buffer suspension without FRET peptide substrate or FRET peptide substrate with buffer only. Fluorescence was measured at 37 °C for 60 min at 2 min intervals at an excitation wavelength of 485 nm and an emission wavelength of 530 nm, using a FLUOstar Omega Microplate Reader (BMG Labtech, Offenburg, Germany). Substrate cleavage was calculated from the fluorescence (F) emitted per minute for each sample. The slope was determined over the time interval from 0 to max 20 min for all FRET substrates. A result was considered positive if the proteolytic activity was at least 5.0 F/min as described previously [8]. Proteolytic activity was ranked as follows: F/min = 5–50, low activity; F/min = 51–500, moderate activity; F/min > 500, high activity. The summated degradation of all FRET peptide substrates examined was used to calculate the total proteolytic activity (TPA). All fecal samples collected were analyzed in triplicate.

To investigate which protease family the measured fecal protease belongs to, the effect of specific inhibitors was studied on a subset of fecal samples in a pilot study. Inhibitors used were benzamidine (BAM), a trypsin and serine protease inhibitor, and ethylenediaminetetraacetic acid (EDTA), an inhibitor of metalloprotease activity (both from Sigma-Aldrich, Amsterdam, The Netherlands). For the inhibitor experiments, 1 mg/ mL feces was incubated with FRET peptide substrate in the presence of 5 mM BAM or 10 mM EDTA. Proteolytic activity was measured as described above.

### 2.4. Statistical Analysis

Statistical analysis of the data was performed using R studio software (R version 4.3.0). The data were checked for normality using the Shapiro–Wilk test with a *p* > 0.05 being normally distributed. For non-normally distributed data, a Mann–Whitney U test was used for unpaired comparisons or a Wilcoxon signed-rank test for paired samples. For normally distributed data, a Welch’s *t*-test was performed. Graphs were generated using the GraphPad Prism 10 software.

## 3. Results

### 3.1. Fecal Sample Collection

A total of 61 fecal samples were analyzed, comprising 26 herbivorous, 23 carnivorous and 12 omnivorous samples. The carnivorous group comprised two large meat eaters, two insectivores and two piscivores. The herbivorous group consisted of two ruminants, two non-ruminants and two frugivores. The characteristics and diets of all animals from which fecal samples were collected are presented in Table 2 and Appendix A.

### 3.2. Proteolytic Profiling of Zoo Animal Fecal Samples

To investigate whether the proteolytic activity differs between animal species with different dietary requirements, the proteolytic activity was measured using a set of 21 fluorescent peptide substrates (Appendix A). Total proteolytic activity (TPA) was significantly higher in feces of carnivores compared to herbivores (*p* < 0.0001) (Figure 1). Also, in feces of the omnivorous animals the TPA was significantly higher than the TPA in feces from the herbivores (*p* = 0.0008) (Figure 1). No significant difference was found in fecal TPA between the carnivorous and omnivorous animals.

Subsequently the TPA levels were compared between animals within the different diet groups. It was observed that the animals with high fecal proteolytic activity were relatively closely related (Figure 2A). The upper part of the phylogenic tree (in gray) consisted only of carnivorous and omnivorous animals, and of these groups, only the pig (*Sus scrofa domesticus*) and slow loris (*Nycticebus pygmaeus*) occurred in the lower part of the tree (Figure 2A). Compared to the other carnivores and omnivores these two animals show a relatively low fecal TPA.

Further analysis of the data showed significant differences between the animals within the three diet groups (Figure 2B–D). In the carnivorous diet group, no differences in proteolytic activity between carnivores, insectivores and piscivores were observed. However, within the carnivore sub-group the fecal TPA of the tiger (*Panthera tigris altaica*) was significantly higher than the lion (*Panthera leo*) (*p* = 0.009). Also, between the two insect eaters a difference was found; fecal TPA of the slow loris was significantly lower than the meerkat (*Suricata suricatta*) (*p* = 0.042) (Figure 2B). Within the omnivore group, pig feces had significantly lower proteolytic activity compared to raccoon (*Procyon lotor*) (*p* = 0.029) and badger (*Meles meles*) (*p* = 0.029) feces (Figure 2C). The animals within the herbivore group were subdivided into ruminants, non-ruminants and frugivores. Comparing the fecal TPA between these subgroups revealed that the non-ruminant animals used in this study had significantly lower fecal proteolytic activity than the ruminant (*p* < 0.0001) and fruit-eating animals (*p* < 0.0001). Within the ruminant group, proteolytic activity was significantly higher in camel (*Camelus bactrianus*) feces compared to giraffe (*Giraffa camelopardalis*) feces (*p* = 0.029) (Figure 2D).

### 3.3. Characterization of Proteolytic Activity in Fecal Samples

To characterize the detected proteolytic activity of animal feces, specific inhibitors were added to the protease assay. The inhibitors used were BAM and EDTA, which inhibit trypsin-like serine protease and metalloprotease activity, respectively. The degradation of the substrates F-R, K-K and R-R by feces from zebra (*Equus grevyi*), giraffe, tiger, meerkat and otter (*Aonyx cinerea*) was inhibited by BAM (Figure 3A–E). Although not all results were statistically significant, an inhibitory trend was observed in all samples. Inhibition was less prominent in degradation of the L-L, F-F and Y-Y substrates. Addition of EDTA to the reaction showed a trend towards a promoting effect on the proteolytic activity for all substrates tested, except for L-L and Y-Y (Figure 4A–E). The degradation of these substrates by the feces of meerkat and otter was slightly inhibited by EDTA.

## 4. Discussion

This study is the first to investigate whether proteolytic activity in feces is related to the diet of herbivores, carnivores and omnivores. Analysis of fecal samples from zoo animals revealed that the carnivores and omnivores included in this study had significantly higher proteolytic activity in their feces than the included herbivores. This is consistent with several comparative studies showing that the production of digestive enzymes varies with diet. Carnivorous mammals generally display higher proteolytic activity in the stomach and pancreas than omnivores and herbivores, reflecting their dependence on protein-rich prey and relatively short gastrointestinal tracts [18,22]. Omnivores exhibit intermediate protease activity that adjusts to the protein content of the diet [23]. In contrast, herbivores are known to have relatively low proteolytic enzyme levels but increased fermentation capacity—an adaptation to carbohydrate- and fiber-rich diets [18]. Significant differences in TPA were also observed between animals within the same diet group. For example, the proteolytic activity in fruit bat (*Rousettus aegyptiacus*) feces was much higher than in lemur (*Lemur catta*) feces, although this observation did not reach statistical significance due to the large variation in TPA of the fruit bat samples. This variation might be related to the large number of animals (*n* = 170) in the enclosure. Despite similarities in diet and genetic relatedness, proteolytic activity was higher in tiger feces than in lion feces. While some variation was observed within the same diet groups, these differences were relatively modest (2- to 10-fold) compared to the much larger differences observed between diet groups (over 100-fold). The higher proteolytic activity found in feces of carnivorous animals compared to herbivorous animal species is consistent with the results of other studies. For example, a study comparing the activities of digestive enzymes in intestinal tissues of omnivorous, herbivorous and carnivorous fish showed that carnivorous fish have higher activities of trypsin and enteropeptidase than herbivorous fish [24]. In addition, humans are known to have significantly less fecal chymotrypsin activity after a short-term vegan diet [25]. This suggests that organisms can adapt to their food intake. The results we observed in pig feces showed a similar result. Although pigs are known to be able to digest meat, at the moment samples were taken at the zoo for this study the pigs were fed a herbivorous diet (Appendix A). This resulted in significantly lower TPA values compared to the other two omnivores that were fed meat like day-old chickens and mice. The variation among omnivores could potentially be linked to the previously mentioned adaptation of chymotrypsin activity in response to the herbivorous diet [18]. Additional experiments with a larger group of omnivores and varying diets are needed to confirm these results.

The observed effect of diet on fecal TPA in pig feces shows that the feeding strategy of the zoos might have influenced the TPA measured in feces in this study. The TPA in the feces in animals living in their natural habitat might therefore significantly differ from animals in captivity.

Another explanation for the differences found in fecal proteolytic activity between the different diet groups could be the variance in intestinal microbiome composition. The microbial diversity is significantly higher in herbivores than in carnivores, while the microbiota in carnivores, unlike herbivores, varies widely within species [26]. Additional experiments, including microbiome analyses, would provide further insight into the role of microbiome composition in the proteolytic activity of feces. The observed difference between the two insectivores might be related to the fact that the slow loris is an exudativore. Which means that, although they are fed some insects, their diet mainly consists of tree gum and other exudates. This is in contrast with the meerkats, which were fed predominantly insects (Appendix A).

From the results with the protease inhibitors, it could be concluded that the proteolytic activity in the feces is probably caused by trypsin-like serine proteases and to a lesser extent by metalloproteinases. This is in line with the observed cleavage patterns during library screening, where substrates containing lysine or arginine residues (K-K, R-R and F-R) were degraded with high efficiency (Appendix A). Trypsin-like serine proteases show a preference for cleaving peptide bonds at lysine and arginine residues [27]. Important trypsin-like serine proteases in food digestion are enteropeptidase, trypsin and chymotrypsin, whose activity is closely related. Enteropeptidases, produced by the small intestine, convert inactive trypsinogen into trypsin, which in turn activates other pancreatic enzymes, such as chymotrypsin and carboxypeptidase A and B [27]. Enteropeptidase, trypsin and carboxypeptidase B have a strong preference for lysine and arginine residues. In turn, chymotrypsin and carboxypeptidase A cleave peptide bonds containing hydrophobic amino acids such as leucine (L), phenylalanine (F) and tyrosine (Y) [28]. This is in line with the observation that proteases in feces from carnivorous and omnivorous animals is able to degrade the hydrophobic substrates F-F, L-L, M-M, W-W and Y-Y with relatively high efficiency (Appendix A). Proteases in feces from herbivorous animals were unable to degrade these substrates. This might be related to the decreased trypsin activity observed in the digestive tract of these animals, resulting in lower activation of chymotrypsin and carboxypeptidase A.

The observed inhibitory effect of EDTA on the degradation of L-L and Y-Y in feces from meerkat and otter suggests that in these samples the substrates L-L and Y-Y are potentially degraded by proteases belonging to the metalloprotease family. The two main metalloproteases present in the digestive tract are carboxypeptidase A and B, the former having a preference for hydrophobic amino acids. Since only two types of inhibitors have been tested, additional experiments with a broader range of protease inhibitors are needed to determine exactly which type(s) of protease(s) are involved.

A limitation of this study is that the number of species and samples is limited. This affects the generalizability of the results. In a follow-up study, we will collaborate with more zoos to include more closely related animals or similar species receiving different feeding strategies. By increasing the group size, we will be able to study the effect of diet on proteolytic activity in feces in more detail.

## 5. Conclusions

The results of this study suggest that diet affects intestinal proteolytic activity. Disturbed proteolytic activity contributes to increased intestinal permeability and thereby to diseases such as gastroenteritis and inflammatory bowel disease [4]. It is possible that changing the diet can alter intestinal proteolytic activity. This needs to be investigated in further experiments with larger cohorts.

## Figures and Tables

**Figure 1 animals-15-03559-f001:**
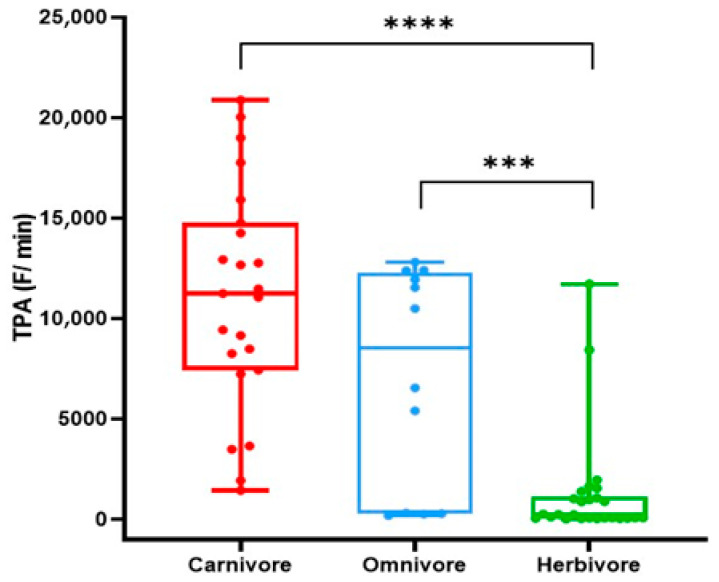
Total fecal proteolytic activity of different diet groups. Comparison of total fecal proteolytic activity (TPA) in F/min in the carnivore, omnivore, and herbivore groups. Unpaired univariate analyses were performed using Mann–Whitney U test. Significance levels are depicted with asterisks: *** *p* < 0.001, **** *p* < 0.0001.

**Figure 2 animals-15-03559-f002:**
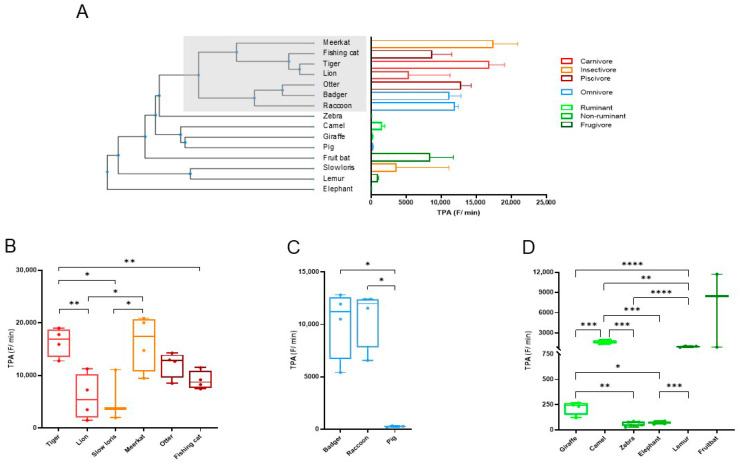
Total proteolytic activity in feces of animals with different dietary requirements. Phylogeny of the 15 animal species analyzed based on TimeTree database [12] and their median fecal total proteolytic activity (TPA) levels in F/min with 95% CI. The gray area depicts the upper part of the tree with animals with high fecal TPA. (**A**). Fecal TPA of the animals investigated divided into the carnivore (**B**), omnivore (**C**) and herbivore (**D**) group. For the omnivorous samples, an unpaired univariate analysis was performed using Mann–Whitney U test, and an unpaired Welch’s *t*-test was performed for the carnivorous and herbivorous samples. Activity was measured in triplicate. Significance levels are depicted with asterisks: * *p* < 0.05, ** *p* < 0.01, *** *p* < 0.001, **** *p* < 0.0001.

**Figure 3 animals-15-03559-f003:**
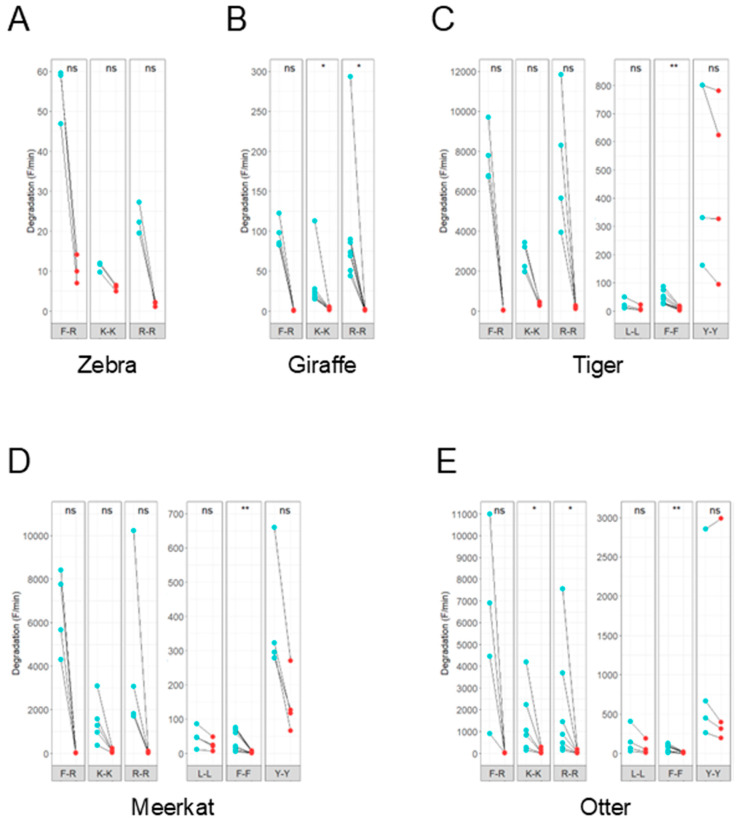
Effect of serine protease inhibitor benzamidine (BAM) on fecal proteolytic activity. Line plots showing the degradation (F/min) of fluorescent peptide substrates before (blue) and after (red) addition of 5 mM BAM. The effect of BAM on the proteolytic activity of feces from animals with different dietary requirements was analyzed: zebra (**A**), giraffe (**B**), tiger (**C**), meerkat (**D**) and otter (**E**). Adding BAM led to a decrease in substrate degradation with varying significance values. A pairwise Wilcoxon signed-rank test was performed. Activity was measured in triplicate. Significance levels are depicted with asterisks: ns: no significant difference, * *p* < 0.05, ** *p* < 0.01.

**Figure 4 animals-15-03559-f004:**
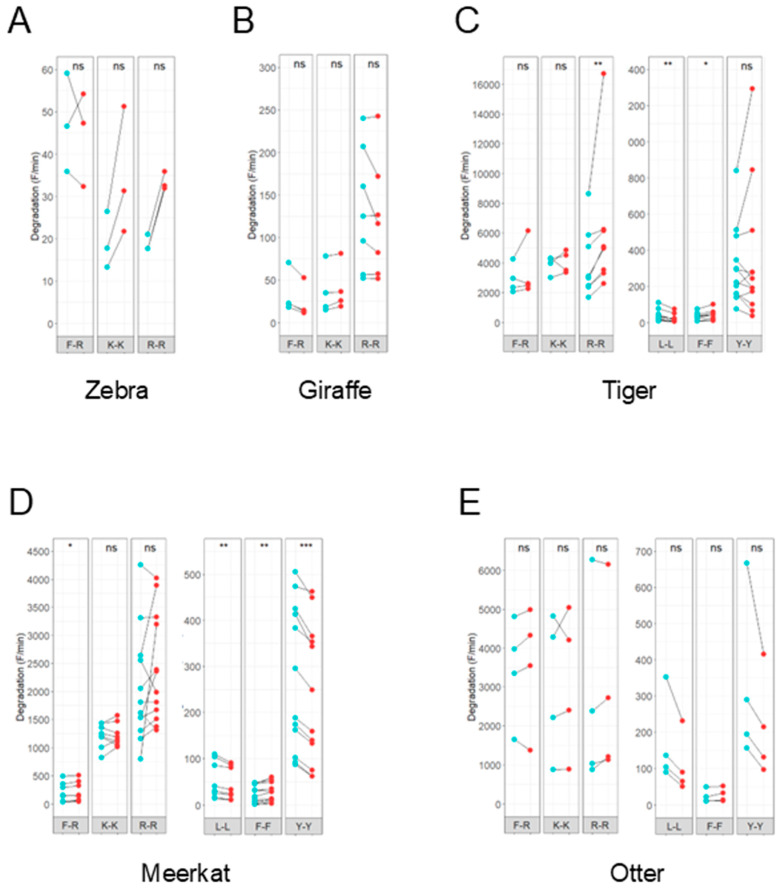
Effect of metalloprotease inhibitor ethylenediaminetetraacetic acid (EDTA) on fecal proteolytic activity. Line plots show the degradation (F/min) of fluorescent peptide substrates before (blue) and after (red) addition of 10 mM EDTA. The effect of EDTA on proteolytic activity of feces from animals with different dietary requirements was analyzed: zebra (*Equus grevyi*) (**A**), giraffe (*Giraffa camelopardalis*) (**B**), tiger (*Panthera tigris altaica*) (**C**), meerkat (*Suricata suricatta*) (**D**) and otter (*Aonyx cinerea*) (**E**). No clear effect upon the addition of EDTA was observed. A pairwise Wilcoxon signed-rank test was performed. Activity was measured in triplicate. Significance levels are depicted with asterisks: ns: no significant difference, * *p* < 0.05, ** *p* < 0.01, *** *p* < 0.001.

**Table 1 animals-15-03559-t001:** Protease substrates used in this study.

Substrate	Sequence
A-A	FITC-Ahx-L-Ala-L-Ala-L-Lys-Dabcyl
C-C	FITC-Ahx-L-Cys-L-Cys-L-Lys-Dabcyl
D-D	FITC-Ahx-L-Asp-L-Asp-L-Lys-Dabcyl
E-E	FITC-Ahx-L-Glu-L-Glu-L-Lys-Dabcyl
F-F	FITC-Ahx-L-Phe-L-Phe-L-Lys-Dabcyl
G-G	FITC-Ahx-L-Gly-L-Gly-L-Lys-Dabcyl
H-H	FITC-Ahx-L-His-L-His-L-Lys-Dabcyl
I-I	FITC-Ahx-L-Ile-L-Ile-L-Lys-Dabcyl
K-K	FITC-Ahx-L-Lys-L-Lys-L-Lys-Dabcyl
L-L	FITC-Ahx-L-Leu-L-Leu-L-Lys-Dabcyl
M-M	FITC-Ahx-L-Met-L-Met-L-Lys-Dabcyl
N-N	FITC-Ahx-L-Asn-L-Asn-L-Lys-Dabcyl
P-P	FITC-Ahx-L-Pro-L-Pro-L-Lys-Dabcyl
Q-Q	FITC-Ahx-L-Gln-L-Gln-L-Lys-Dabcyl
F-R	FITC-Ahx-L-Phe-L-Arg-L-Lys-Dabcyl
R-R	FITC-Ahx-L-Arg-L-Arg-L-Lys-Dabcyl
S-S	FITC-Ahx-L-Ser-L-Ser-L-Lys-Dabcyl
T-T	FITC-Ahx-L-Thr-L-Thr-L-Lys-Dabcyl
V-V	FITC-Ahx-L-Val-L-Val-L-Lys-Dabcyl

**Table 2 animals-15-03559-t002:** Characteristics of the animals from which fecal samples were collected in this study.

	Animal Species	No. of Animals in the Enclosure	No. of Fecal Samples Collected	Subgroup ^†^
Carnivore	Tiger (*Panthera tigris altaica*)	2	4	n.a.
	Lion (*Panthera leo*)	3	4	n.a
	Slow loris (*Nycticebus pygmaeus*)	4	3	Insectivore
	Meerkat (*Suricata suricatta*)	7	4	Insectivore
	Asian small-clawed otter (*Aonyx cinerea*)	4	4	Piscivore
	Fishing cat (*Prionailurus viverrinus*)	2	4	Piscivore
Omnivore	Raccoon (*Procyon lotor*)	6	4	n.a.
	Badger (*Meles meles*)	2	4	n.a.
	Bentheim black pied pig (*Sus scrofa domesticus*)	2	4	n.a.
Herbivore	Giraffe (*Giraffa camelopardalis*)	4	4	Ruminant/ Fore-gut fermenter
	Camel (*Camelus bactrianus*)	3	4	Ruminant/ Fore-gut fermenter
	Zebra (*Equus grevyi*)	3	7	Non-ruminant/ Hind-gut fermenter
	Elephant (*Elephas maximus*)	5	4	Non-ruminant/Hind-gut fermenter
	Ring-tailed lemur (*Lemur catta*)	11	4	Frugivore
	Fruit bat (*Rousettus aegyptiacus*)	170	3	Frugivore

^†^ n.a.: not applicable.

## Data Availability

Data are available upon reasoned request.

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
