# Peer review of "Distinct Fecal Proteolytic Activity in Zoo Animals with Different Feeding Strategies"

_animals, 2025, doi:10.3390/ani15243559_

Round 1
Reviewer 1 Report
Comments and Suggestions for Authors
My Review

Reviewer 2 Report
Comments and Suggestions for Authors
The manuscript is devoted to an important and relevant question: are there any differences in feces proteolytic activity across animals using different feeding strategies. The research is well explained, and uses an experimental approach. The results show higher protease activity in carnivores and omnivores compared to herbivores. The research itself important for understanding digestive physiology and for design of non-invasive markers for digestive system functional activity.
There are some weaknesses limiting the work. The Introduction and discussion are long and do not emphasize the knowledge gap. Figures and results are very detailed and sometimes overcrowded with numbers. The reference list is really short. Only 14 citations. Please, work on introduction and discussion, diverse referencing and widen the list. The self-citation rate is high. So authors really need to work with the text.
Simple summary:
Please, remove technical terms, or explain them. Please, add a sentence explaining why the research was conducted. Add practical implications in the end.
Abstract:
Rewrite the text so that it follows imrad structure. Now it jumps between topics and reads like a condensed abstract.
Introduction:
The text is long and dense, describing different types of diet. But less focus is on the current knowledge gap. Correct it. Cite more references, the text would benefit from it. State the aim of research earlier and more precise.
Methods
The section is quite detailed. However, clarify these several points.
Add an ethical statement.
Only 2 species per subgroup were analysed- it’s difficult to generalise the results. Mention this limitation in discussion.
The group of omnivores is underrepresented.
Were the feces freshly collected (what time since defecation)?
The samples from different individuals were analysed separately?
Protease activity is sensible to freeze-melting. Were these processes avoided?
Please, justify criteria for high protease activity.
The authors tested only 2 inhibitors. Please, mention this limitation in discussion.
Did you apply multiple comparisons corrections?
How did you handle triplicate measurements?
Did you account for interindividual variability?
Results:
The section is overcrowded with numbers. Please, move them into a table or graph.
Figures 1 and 2 are underinterpreted. Please, describe any trend if it’s possible. Results also jump between group and individual comparisons without clear structure. Please, correct it.
Figures: clarify axis labels and explain abbreviations and asterisks in figure captions.
Discussion
Rewrite the section expanding the reference list and reduce self-citation. Write the text with hierarchy. Now it jumps from topic to topic like it does in introduction and results. Follow the structure: differences-exceptions-explain the difference- provide possible implications-discuss limitations-provide future directions.
Conclusions:
Now it restates what is already present in results.
Highlight the novelty and importance. Mention limitations and add future directions.
Reviewer 3 Report
Comments and Suggestions for Authors
See attachment.

Round 2
Reviewer 1 Report
Comments and Suggestions for Authors
good
Author Response
Dear reviewer,
Thank you again for reviewing!
We agree that the research design is not optimal, this is related to the fact that we collaborated with only two zoos with limited access to species. Therefore we added the suggestions to collaborate with more zoos for future research in the revised version submitted Oct 31st.
Kind regards,
Wendy
Reviewer 2 Report
Comments and Suggestions for Authors
The version 2 is a substantial improvement over version1. The text is better structured. Most of the comments are met by the authors.
However, the reference list is still too short, only 16 references, which is not suitable for a manuscript. Please, expand the reference list, add recent 2020-2025 citations, and provide comparative analysis of your research with recent data. Rewrite or add missing information from recent manuscripts in introduction and discussion.
Please, reorganise figures 3 and 4. The graphs are too small.
Author Response
We would like to thank the reviewer for his/ her compliments on the improvement of our manuscript. We totally agree!
The amount of references in the manuscript is indeed low therefore we re-searched Pubmed for comparative studies and found the use of protease inhibitors in the treatment of intestinal diseases. Text on this topic was added to the introduction section (Marked yellow, Line 51-60) and this led to lengthening of the reference list with 4 articles. In addition we found one recent review on the digestive system. In the end 5 additional references were added to the manuscript (Added references are marked in yellow).
Based on the suggestion of the reviewer we increased the size of Figure 3 and 4. We agree that they were relatively small.
We hope that these adjustments are sufficient to accept the manuscript for publication in Animals.
Kind regards,
Wendy Kaman
Reviewer 3 Report
Comments and Suggestions for Authors
N/A
Author Response
Dear reviewer,
Thank you again for reviewing!
Within the version resubmitted on Oct 31st we added additional information on the methods used to clarify the execution of the described research.
Kind regards,
Wendy